# Robot Grasping Based on Stacked Object Classification Network and Grasping Order Planning

**Chenlu Liu** [1], **Di Jiang** [1], **Weiyang Lin** [1,2,*] and **Luis Gomes** [3,*]

1   Research Institute of Intelligent Control and Systems, Harbin Institute of Technology, Harbin 150001, China; chenluliu.cll@gmail.com (C.L.); jiangdi1998@icloud.com (D.J.)
2   Ningbo Institute of Intelligent Equipment Technology Company Ltd., Ningbo 315200, China
3   Centre of Technology and Systems, NOVA School of Sciences and Technology, NOVA University Lisbon/UNINOVA, 2829-516 Monte de Caparica, Portugal
*   Correspondence: wylin@hit.edu.cn (W.L.); lugo@fct.unl.pt (L.G.)

**Abstract:** In this paper, the robot grasping for stacked objects is studied based on object detection and grasping order planning. Firstly, a novel stacked object classification network (SOCN) is proposed to realize stacked object recognition. The network takes into account the visible volume of the objects to further adjust its inverse density parameters, which makes the training process faster and smoother. At the same time, SOCN adopts the transformer architecture and has a self-attention mechanism for feature learning. Subsequently, a grasping order planning method is investigated, which depends on the security score and extracts the geometric relations and dependencies between stacked objects, it calculates the security score based on object relation, classification, and size. The proposed method is evaluated by using a depth camera and a UR-10 robot to complete grasping tasks. The results show that our method has high accuracy for stacked object classification, and the grasping order effectively and successfully executes safely.

**Keywords:** robot grasping; stacked object classification; grasping order planning

## 1. Introduction

ROBOTS are expected to perceive objects, plan action sequences, and complete tasks independently when they are manipulating in complex and diverse environments. Particularly in grasping tasks, a robot needs to sense the surrounding information so that it can plan reasonable grasp action order according to the environmental constraints and finish its grasping tasks.

There are two main steps in a robot grasping task: object detection and grasp order planning [1]. Through object detection, the robot can perceive the environment and obtain information about the objects [2–4]. Following the grasping order planning, the robot finds appropriate transit and transfer motions to move a set of objects and avoid collisions with the environment [3].

Numerous studies have been conducted on object detection and grasping order planning in industrial robots, and there are some key points in grasping tasks: (1) A stable recognition algorithm is essential. For the classification of stacked objects, a great challenge is that different occlusion will produce many different or even strange shapes because of the variety of stacked scenarios It requires that the recognition algorithm should accurately detect the objects and obtain their geometric features. (2) For stacked objects, it is important to extract geometric relations and dependencies between entities in the scene, so that the robot can plan the motion to avoid object falling or collision issues.

Focusing on these two points, a robot grasping approach based on SOCN and grasping order planning method for stacked objects is proposed in this paper. The architecture of the proposed approach is shown in Figure 1. First, The point cloud data of objects is obtained and segmented as the input of our method. Then using the proposed SOCN and relationship exploration network, we can get the object classification and relationship between objects.

Finally, by calculating the security score of object classification, relationship, and size, the grasping order is planned to complete the grasping task.

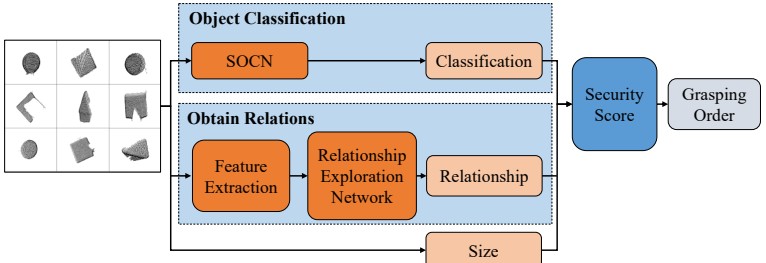

**Figure 1.** Architecture of the proposed robot grasping approach.

The contributions of our work can be summarized as follows:

(1) A stacked object classification network (SOCN) is proposed for stacked object recognition. The SOCN adopts the self-attention mechanism to conduct feature learning and takes the spatial scale into account to further adjust the inverse density, which makes it achieve impressive recognition accuracy for stacked object classification.

(2) Proposed a safe and effective grasping order planning method for stacked objects. The method can automatically obtain the geometric relations and dependencies in stacked objects by using the relationship exploration network based on visual sensor data. Different from the method that only considers the relations between objects, the method considers the object relations, object class, and object size to calculate the security score so as to plan the grasping order, which makes it both safe and flexible in diverse object grasping tasks.

This paper is organized as follows. Section 2 introduces the related work about grasping tasks. Section 3 presents the algorithm to realize point cloud segmentation and the proposed SOCN to achieve point objects classification. Section 4 presents the grasping order planning method based on security scores. In Section 5, robot grasping experiments are carried out, including objects classification experiments and grasping order experiments. The conclusion is provided in Section 6.

## 2. Related Work

### 2.1. Object Classification and Detection

To accomplish a grasping task, the ability of environment perception and object detection is increasingly integrated into robot control. To address this, a great deal of research work is developed. In [5], the author designed a human-robot interaction collaborative workstations, in which the robot can complete speech recognition and object detection, so as to realize object grasping. Li et al. use QR codes to obtain information about objects. By identifying the QR code, the category and location of the object are determined for robot grasping [6]. Wang, Panda et al. design an improved SIFT (Scale Invariant Feature Transform) algorithm to realize object segmentation and localization [7,8]. Although these methods are feasible in certain cases, they depend on image features and are sensitive to image noise [9]. To satisfy the requirements of these methods, the robot operating workspace needs to be strictly controlled, hence these methods are not generally applicable. However, in stacked object grasping tasks, image features used in these methods may be hidden due to the occlusion issue. So far extensive studies focus on object detection by using the deep neural networks [10]. In works [11–13], researchers use YOLO as the backbone feature extractor and realize the object detection. Furthermore, Shi et al. introduce Light-Head RCNN (Region Convolutional Neural Network) into the Mask-RCNN network to improve the speed of object detection [14]. Despite the above algorithms having good performance in object detection, these algorithms have shortcomings in estimating the object geometric information, like geometric size and posture.

Recently, some depth cameras such as Intel RealSense, Microsoft Kinect, etc. are employed to perceive 3D scenes. In most cases, these sensors generate 3D images as point

cloud data and thus they describe 3-D information of stacked objects. Lin et al. use RGB-D cameras to obtain depth images and implement object pose estimation [15]. In [16], the author uses only the depth information and its geometric features to segment each object and estimate the position and orientation in a stacked scene. In [17], the author constructed a conditional random field to model the semantic contents in stacking object regions, and it helps the robot achieve a 69.4% success rate for task-oriented grasping. Qi et al. propose a pioneering deep learning method called PointNet for 3D classification and segmentation [18]. Wang et al. adapted an improved network named Q-PointNet to achieve intelligent stacked object grasping using an RGBD sensor [19]. Furthermore, Wu et al. present a novel network called PointConv to perform convolution on 3D point clouds and achieve better segmentation results [20]. It uses an inverse density scale to re-weight the continuous function learned by multi-layer perception (MLP) and outperforms the prior works on challenging semantic segmentation benchmarks for 3D point clouds.

On the other hand, recent studies show that transformer architecture significantly improves the performance on various tasks and demonstrates high accuracy and real-time properties on object detection [21–23]. Its self-attention mechanism is effective in modeling the interactions without regard to spatial variance at both inputs and outputs.

In this paper, we propose a novel deep-learning-based SOCN which includes a transformer architecture for stacked objects detection. The proposed network employs the visible volume of objects to further adjust the inverse density parameters and has the self-attention mechanism to improve the accuracy of object detection for stacked objects.

## 2.2. Grasp Strategy

For stacked object grasping, robots need to formulate grasp strategies and carry out reasonable grasping orders to ensure safety. Guo et al. proposed a shared CNN (Convolutional Neural Network) to implement object discovery and grasp detection in stacked scenes [24]. However, this work only picks up the most exposed fruit from a stack of fruits and lacks overall planning for grabbing stacked hidden objects. Zhang et al. conduct semantic understanding of objects using the two-stage CNN and construct relational structure object tree [25,26]. The input of their method is 2D images, and they accomplished the corresponding grasping work with a success rate of 59.4% in stacking scenes. In comparison to these methods, our method employs 3D point clouds as input to reduce the interference of light variation. In addition to this, our method considers both physically support relations and object classification to plan the grasping order.

Another work [27] proposes a deep reinforcement learning approach to grasp occluded objects. It uses the Q-learning method to learn a target-oriented motion critic. This method pushes down and spreads out cluttered objects to rearrange them, which is not suitable for fragile objects. In [11,28], Markus et al. propose graph-based semantic perception and physically plausible support relations extraction method. However, these methods only give the relationship between different objects, and do not provide the overall order planning of object grasping. At the same time, these methods are not applicable in the case of a single manipulator for grasping stacked objects. In our paper, the relationship between objects is extracted using the designed relationship exploration network, and a grasping order planning method for stacked objects by using a single manipulator is presented.

## 3. Stacked Objects Detection: Point Cloud Segmentation and Objects Classification

In this section, a stacked object classification network SOCN is presented to classify stacked objects. There are two steps in the method. First, the point cloud data of stacked objects are segmented. Then the module PointConvSSN in SOCN is employed to extract the features of the segmented point clouds and the module transformerSSN is employed to realize the object classification, where SSN is the spatial scale in the network.

### 3.1. Point Cloud Segmentation

For stacked objects, a robust point cloud segmentation is crucial to separate the objects and obtain the hierarchical relationship [29].

In this subsection, the point cloud segmentation is realized to make each object independent as shown in Figure 2. The complexity of robot operating space affects the search speed of stacked objects. Especially, the more objects in the robot operating space, the slower the search becomes. Therefore, the passthrough algorithm [30] is employed to determine the spatial range and remove irrelevant objects. Besides, a model-fitting approach random sample consensus (RANSAC) [31] is used to remove the platform point cloud data. Due to the fitting error between the plane model and the actual scanning plane, noise artifacts appear between the object and the removed plane. To eliminate these artifacts, the average distance $d$ between point $p_0(x, y, z)$ and other points $p_i(x_i, y_i, z_i)$ is calculated using Equations (1) and (2) in the neighborhood $U(x, y, z, \varepsilon)$ of $p_0$, where $\varepsilon$ is the range threshold, $n$ is the amount of points. If $d$ exceeds the distance threshold $d_{thre}$, the point $p$ is regarded as an outlier and removed from the point clouds.

$$U = \{p | |p - p_0| < \varepsilon\} \tag{1}$$

$$d = \frac{1}{n} \sum_{i=1}^{n} \sqrt{(x - x_i)^2 + (y - y_i)^2 + (z - z_i)^2} \tag{2}$$

After the above processing, the stacked objects are divided by the locally convex connected patches (LCCP) algorithm [32]. Algorithm 1 summarizes the details of the point cloud segmentation methodology.

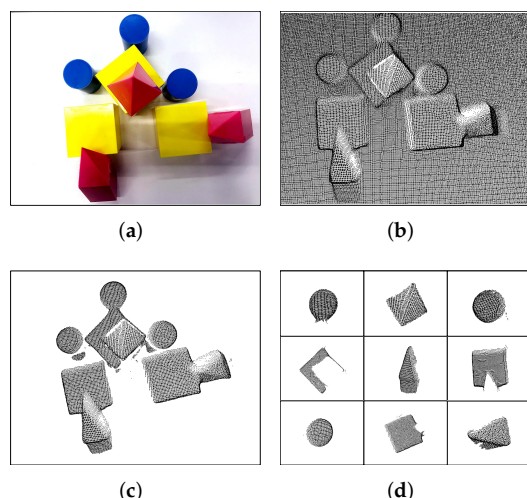

(a)

(b)

(c)

(d)

**Figure 2.** Point Cloud segmentation of stacked objects. (**a**) the object appearance seen from RGB camera. (**b**) the original point cloud read from depth camera. (**c**) pre-processes point cloud via passthough filter, RANSAC and outlier removal. (**d**) the point cloud segmented by LCCP.

---

**Algorithm 1** Point Cloud Segmentation

---

**Input:** Initial point cloud *pc*
**Output:** point clouds after segmentation *pcs*
  1: **if** *pc* != *null* **then**
  2:      pc_pass ← Passthrough(input, pass_threshold)
  3:      pc_ransac ← RANSAC(pc_pass, plane_threshold)
  4:      pc_outlier ← $d$(pc_ransac, $U(x, y, z, \delta, \varepsilon, \gamma)$) < $d_{thre}$
  5: **end if**
  6: pcs_lccp ← LCCP(pc_outlier)
  7: N ← count the number of objects after segmentation
  8: **for** $i = 0$ to N **do**
  9:      height ← calculate average height (pcs_lccp[i])
 10:      size ← calculate the number of points (pcs_lccp[i])
 11:      **if** height $< height\_thre$ **and** size $> size\_thre$ **then**
 12:          pcs[i] ← pcs_lccp[i]
 13:      **end if**
 14: **end for**

---

### 3.2. Stacked Objects Classification

Figure 3 shows the structure of the stacked object classification network. The input of the network is the point cloud pre-processed in Section 3.1, and the output is the object classification. The entire network can be split into two modules:

- A spatial features network PointConvSSN that considers the spatial scale is proposed to extract the features of the stacked object point cloud.
- A transformerSSN is designed to realize the classification of objects. Its self-attention mechanism carries out soft-search over the input features of stacked objects and classifies objects based on the important features.

#### 3.2.1. Pointconvssn

As shown in Figure 4, points are un-ordered and thus they do not conform to the regular grids as in 2D images. Because the weight function of convolutional neural networks highly depends upon the distribution of input point cloud [20], PointConv takes non-uniform sampling into account and uses inverse density $S(p')$ on the learned weights to compensate for the non-uniform sampling. The convolution of PointConv is defined as:

$$\sum_{(\delta_x, \delta_y, \delta_z) \in G} S(p)W(p)F(x + \delta_x, y + \delta_y, z + \delta_z) \tag{3}$$

where $F(x + \delta_x, y + \delta_y, z + \delta_z)$ is the feature of point $p(x, y, z)$ in the local region $G$ centered around the point $p$. $S(p)$ is the inverse density at point $p(\delta_x, \delta_y, \delta_z)$ and $W(p)$ is a weight function approximated by a multi-layered perception (MLP).

However, for stacked objects, the density of local point clouds changes due to object occlusion, as shown in Figure 4c. Intuitively, if there is no occlusion, as in Figure 4b, the contribution of points, which are very close to one another in the local region like $p_1, p_2, p_3, p_4, p_5, p_6$, should be smaller. If occlusion exists, the corresponding contribution should be further adjusted because of the distribution of the occluded point cloud has changed.

Our main contribution, in this case, is to consider the spatial scale which is the visible volume of the stacked object denoted as $V$. It indicates the occlusion degree of stacked objects and it is a global feature for points in the same point cloud, hence $S(p)$ can be further adjusted according to the $V$ and the network can better approximate the 3D convolution on the point cloud of stacked objects.

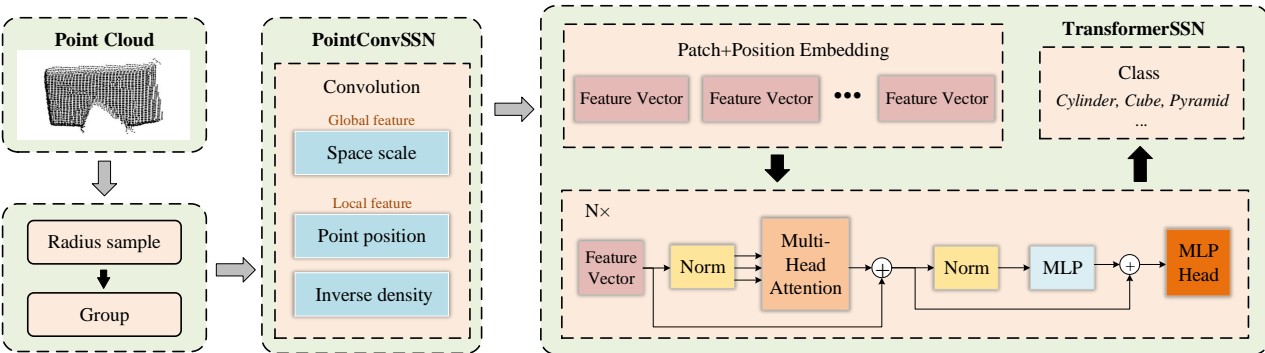

**Figure 3.** Structure of the Stacked Object Classification Network (SOCN). The segmented point clouds enter PointConvSSN after radius sampling and grouping. Then, the feature vectors obtained by 3D convolution are sent to TransformerSSN for extraction of more semantic information, and finally the object classification is obtained.

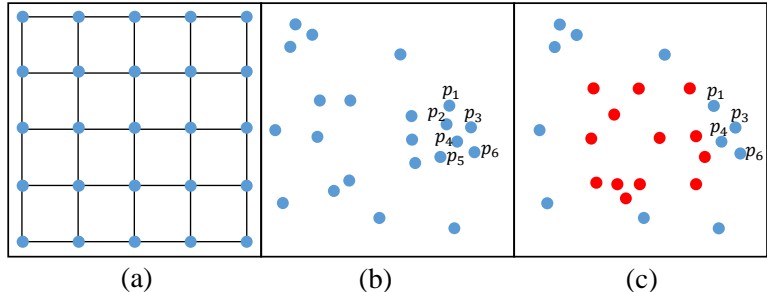

(a)          (b)          (c)

**Figure 4.** Image grid, point cloud and stacked object point cloud (**a**) shows a $5 \times 5$ local region in a 2D image, where distance between points are constant; (**b**) shows a local region of a point cloud, the order and relative positions of the points are non-uniform. (**c**) shows a local region of a stacked point cloud. The blue one is obscured by the red one.

Therefore, the volume of each point cloud is first calculated and taken as the volume parameter of each point in this point cloud. Assuming $\mathbf{x} = \{x_1, x_2, \ldots, x_n\}$, $\mathbf{y} = \{y_1, y_2, \ldots, y_n\}$ and $\mathbf{z} = \{z_1, z_2, \ldots, z_n\}$, where $x_i, y_i, z_i$ are coordinate of point $p_i(x_i, y_i, z_i)$ with $n$ amount in a point cloud. To calculate the visible volume of objects, the point cloud is cut into slices of equal thickness $dz$ along the $z$-axis and each slice is divided into meshes $ds$ of the unit size. Then the grids are integrated to estimate the area of each slice and further integrated along the $z$-axis for an estimated volume of the objects. The formulation is expressed as:

$$V(p) = \int dv = \iint 1 ds dz \tag{4}$$

Subsequently, the spatial scale and inverse density are attributed to each point and fed into the MLP, shown as Figure 5. Because spatial scale is considered, therefore we call this network PointConvSSN (SSN stands for spatial scale network). The convolution in this network is defined as follows:

$$\sum_{(\delta_x, \delta_y, \delta_z) \in G} V(p)^{-1} S(p) W(p) F(x + \delta_x, y + \delta_y, z + \delta_z) \tag{5}$$

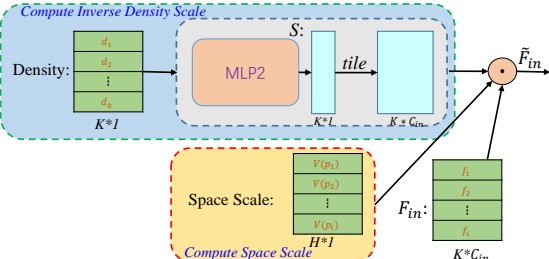

**Figure 5.** Part of PointConvSSN The blue area is the inverse density part of the PointConv and the yellow part is the spatial scale which is employed to estimate the influence of object occlusion on inverse density.

3.2.2. Transformerssn

The Transformer's attention mechanism enables it to achieve better performance on objects classification tasks. To better learn the extracted features by PointConvSSN, the spatial transformer (TransformerSSN) is designed as shown in Figure 3.

We divide the PointConvSSN output feature vector $\mathbf{q}$ into $N$ patches of fixed size, embed each patch linearly, and feed back the vector sequence to an encoder for a 3D point cloud. The encoder is composed of a multi-headed self-attention layer and an MLP layer. Layernorm (LN) is added before each layer, and residual connections after each layer. A Relu function is added between every two layers of MLP. Finally, the MLP header is obtained by decoding, which represents the classification Class of the point cloud. The formulation is summarized as:

$$\mathbf{a}_0 = [\mathbf{q}_{\text{class}}; \mathbf{q}^1\mathbf{M}; \mathbf{q}^2\mathbf{M}; \cdots ; \mathbf{q}^N\mathbf{M}] + \mathbf{M}_{\text{pos}} \tag{6}$$

$$\mathbf{a}'_l = \text{MSA}(\text{LN}(\mathbf{a}_{l-1})) + \mathbf{a}_{l-1}, l = 1 \ldots L \tag{7}$$

$$\mathbf{a}_l = \text{MLP}(\text{LN}(\mathbf{a}'_l)) + \mathbf{a}'_l, l = 1 \ldots L \tag{8}$$

$$\text{Class} = \text{LN}(\mathbf{a}_L^0) \tag{9}$$

where $\mathbf{M}$ is the block of feature weight and $\mathbf{M}_{\text{pos}}$ is the position weight.

## 4. Grasping Order Planning

For stacking objects, improper grasp order will cause stacked objects to drop. The position and size of objects are the factors that affect the stability of stacked objects. Meanwhile, the classification and relationship between objects are also important factors in grasping order planning. In this section, a grasping order planning method is presented which considers the object relationship, size, and classification to obtain a reasonable order for the grasping tasks.

### 4.1. Spatial Feature Obtain

The geometric relations and dependencies of stacked objects are affected by the attitude and height of the object. At the same time, the overlapping degree of stacked objects also affects the object relationship. Therefore, spatial features of stacked objects including the normal vector of the top surface of the bounding box, their average height, and the overlapping degree of stacked objects are extracted.

The principal component analysis (PCA) method is used to obtain eigenvectors of the object [15], and then the normal vector of object top surface is selected as the object attitude representation. Denoting $\mathbf{U}$ as an orthogonal matrix and $\mathbf{n}$ as the normal vector, where $\mathbf{n}$ is the column element of matrix $\mathbf{U}$ and $\mathbf{n}$ is perpendicular to the object upper surface, the normal vector is obtained using Equations (10) and (11).

$$\mathbf{A} = \mathbf{U}^{-1} \Lambda \mathbf{U} \tag{10}$$

$$\mathbf{A} = \begin{bmatrix} \mathrm{cov}(\mathbf{x}, \mathbf{x}) & \mathrm{cov}(\mathbf{x}, \mathbf{y}) & \mathrm{cov}(\mathbf{x}, \mathbf{z}) \\ \mathrm{cov}(\mathbf{y}, \mathbf{x}) & \mathrm{cov}(\mathbf{y}, \mathbf{y}) & \mathrm{cov}(\mathbf{y}, \mathbf{z}) \\ \mathrm{cov}(\mathbf{z}, \mathbf{x}) & \mathrm{cov}(\mathbf{z}, \mathbf{y}) & \mathrm{cov}(\mathbf{z}, \mathbf{z}) \end{bmatrix} \tag{11}$$

Meanwhile, the average height $h$ of the objects is calculated using Equation (12).

$$h = \frac{1}{n} \sum_{i=0}^{n} z_i \tag{12}$$

The overlapping degree of stacked objects determines the position of the objects and the safety of grasping. The intersection area of the minimum contour rectangle of the point cloud is employed to calculate the overlapping degree $\gamma_j$, shown as Equation (14).

$$S_{\mathrm{over}} = S_a \cap S_b \tag{13}$$

$$\gamma_j = S_{\mathrm{over}} / S_j, j = a, b \tag{14}$$

where $S_a$ and $S_b$ are minimum circumscribed rectangle areas observed from a fixed overlooking perspective.

*4.2. Relationship Exploration*

To obtain the relationship across stacked objects, a fully connected neural network is constructed, as shown in Figure 6. The normal vector $\mathbf{n}_a$, $\mathbf{n}_b$, average height $h_a$, $h_b$ and overlapping degree $\gamma_a$, $\gamma_b$ of the object $O_A$ and $O_B$ are calculated as the input $\mathbf{u}$ of the neural network. The relationship prediction is the output. The relationship of objects is divided into five categories: $O_A$ supports $O_B$(UD), $O_B$ supports $O_A$(DU), $O_A$ leans against $O_B$(LS), $O_B$ leans against $O_A$(SL), no relevancy between $O_A$ and $O_B$(NR). The set Re is defined as:

$$\mathrm{Re}(O_A, O_B) \in \{\mathrm{UD}, \mathrm{DU}, \mathrm{LS}, \mathrm{SL}, \mathrm{NR}\} \tag{15}$$

Define the labels for these five relationships as:

$$\lambda(x) = \begin{cases} 1, & x = \mathrm{UD} \\ 2, & x = \mathrm{DU} \\ 3, & x = \mathrm{LS} \\ 4, & x = \mathrm{SL} \\ 5, & x = \mathrm{NR} \end{cases} \tag{16}$$

The constructed network has an input layer, two hidden layers, and an output layer. The Relu is employed as a hidden layer activation function and Softmax is employed as an output layer activation function. The details of the Softmax function in our network is shown as Equation (17):

$$P_i(\lambda(\mathrm{Re}(O_A, O_B)) = i | \mathbf{u}) = \frac{e^{y_i}}{\sum\limits_{i=1}^{n} e^{y_i}}, n = 5 \tag{17}$$

where $P_i$ is the probability, and $y_i$ is the prediction value of the network. Finally, we utilize NLLLoss multi-category cross entropy function to calculate the loss of expectation, shown as Equation (18):

$$Loss(\mathbf{u}) = -\sum_{i}^{n} t_i ln(P_i), n = 5 \tag{18}$$

where $t_i$ is the true value and it is regarded as 1 when the $\lambda(\mathrm{Re}(O_A, O_B)) = i$, otherwise it is 0. To accelerate convergence and improve training accuracy, the network applies the stochastic gradient descent method with momentum during the training stage.

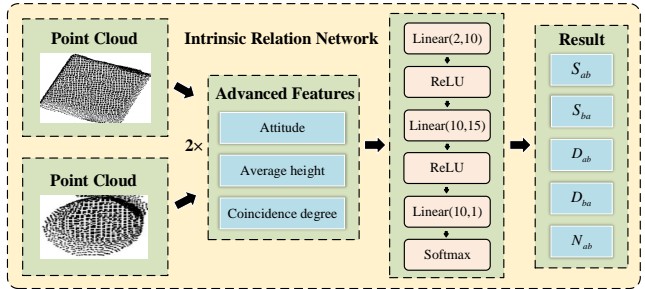

**Figure 6.** Structure of the Relationship Exploration Network.

*4.3. Grasping Order Planning Based On Security Mechanism*

According to prior knowledge, a grasping order planning method is proposed based on a security mechanism that calculates the security score of relationship, classification, and size. The proposed grasping order planning method is composed of three steps: outlier objects extraction, hierarchical statistics, attribute subdivision.

Before presenting the method, we employ $Q_A$ as the stacked objects set containing $n$ amount of stacked objects,

$$Q_A = \{a_1, a_2, \ldots, a_n\} \tag{19}$$

*Outlier objects extraction:* Assuming $Q_B$ as the dense and contiguous object set and $Q_C$ as the discrete object set, $Q_B$ and $Q_C$ are obtained by

$$Q_B = \{a \in Q_A \mid \mathrm{Re}(a, b) \neq , \exists b \in Q_A, b \neq a\} \tag{20}$$

$$Q_C = Q_A - Q_B \tag{21}$$

*Hierarchical statistics:* Set the objects above other objects as set $Q_D$

$$Q_D = \{a \in Q_B \mid \mathrm{Re}(a, b) = \mathrm{UD} \\ \text{or } \mathrm{Re}(a, b) = \mathrm{LS}, \exists b \in Q_B, b \neq a\} \tag{22}$$

The objects below are represented as set $Q_E$.

$$Q_E = Q_B - Q_D \tag{23}$$

After obtaining the sets $Q_C$, $Q_D$ and $Q_E$, we can employ them to evaluate the relation security score $\psi$ of stacked objects relationship.

$$\psi(a) = \begin{cases} \mathrm{RS}_{\text{descrete}}, & a \in Q_C \\ \mathrm{RS}_{\text{top}}, & a \in Q_D \\ \mathrm{RS}_{\text{bottom}}, & a \in Q_E \end{cases} \tag{24}$$

*Attribute subdivision:* The class and size of stacked objects are also important to planning the grasp order. According to the categories of objects, the set $Q_A$ is divided into $C_1, C_2, \cdots, C_m$, where $C_i \subseteq Q_A$ and $\forall C_i, C_j, C_i \cap C_j = \varnothing$. Similarly, the set $Q_A$ is divided into three subsets $S_s, S_m$, and $S_l$ by classifying objects into three categories small, medium, and large based on their size. Then the class security score $\phi(a)$ and size security score $\omega(a)$ are obtained by:

$$\phi(a) = \begin{cases} \mathrm{CS}_1, & a \in C_1 \\ \mathrm{CS}_2, & a \in C_2 \\ \quad \vdots \\ \mathrm{CS}_m, & a \in C_m \end{cases} \tag{25}$$

$$\omega(a) = \begin{cases} SS_s, & a \in S_s \\ SS_m, & a \in S_m \\ SS_l, & a \in S_l \end{cases} \tag{26}$$

Since the category of stacked objects is considered, the grasping planning order can be carried out for different stacked objects. Finally, the total security score $\delta$ can be calculated by sum of $\phi$, $\psi$ and $\omega$

$$\delta(a) = \psi(a) + \phi(a) + \omega(a) \tag{27}$$

The Grasping Order (GO) $(o_1, o_2, \ldots, o_n)$ is determined by the following condition,

$$\delta(o_1) > \delta(o_2) > \ldots > \delta(o_{n-1}) > \delta(o_n) \tag{28}$$

where $o_i \in Q_A$.

## 5. Experiments

In this section, the stacked objects detection and grasping order planning method are evaluated. We present the performance of the proposed SOCN compared with some state-of-the-art methods in stacked objects classification scenes and test the grasping order planning method in the real grasping system. The experiment video is available in the Supplementary Materials.

### 5.1. Data Collection and Preprocessing

By using a depth camera Azure Kinect DK, we collected a large number of point cloud data of stacked scenes, which contains as many different object stacking situations as possible. There are three classes of objects in each scene: cylinders, pyramids, and cubes. Using the point cloud segmentation method presented in this paper, the point cloud data is segmented and more than 500 different point cloud data of objects are obtained in each class (see Figure 2). We use software and manual annotation to mark objects of different shapes. As a result, more than 1500 object point cloud data are gained for training and testing the model. A RTX 2080Ti GPU (GIGIABYTE, Shenzhen, China) is employed to train and test the network. Further, to evaluate the proposed SOCN and grasping order planning method, the camera was fixed on the bracket and a 6-DoF robot UR-10 is employed to perform the experiments (see Figure 7).

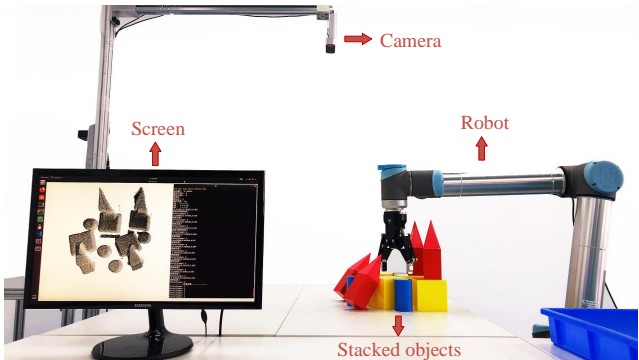

**Figure 7.** Grasping Experimental System. An Azure Kinect DK camera is employed to obtain the image. A UR-10 robot is employed to grasp the stacked objects.

### 5.2. Stacked Object Classification Results and Analysis

Two experiments for stacking object classification are designed for comparison. In the first experiment, we evaluate the classification performance of SOCN and PointConv for each segmented single object point cloud data PointConv is a state-of-the-art object point cloud classification method [20] and it has been selected for our comparison experiment. In the second experiment, the classification performance of SOCN in the actual stacked object

scene is evaluated. Different objects including cylinders, pyramids, and cubes are placed randomly and stacked on the workbench.

### 5.2.1. Segmented Single Object Classification

A training set containing 1200 object point cloud data samples is constructed to train the two networks and 300 samples are employed to test the model. Figure 8 shows the variation of classification accuracy with the number of epochs in the training process of the two networks. It can be seen that with the increase of epochs, SOCN achieves the satisfying accuracy faster and smoother than PointConv. At the same time, the classification accuracy in the test dataset is shown in Table 1. The result shows that the classification performance of SOCN is up to 90.3%, which demonstrates that spatial scale and transformer in SOCN are effectively improving the performance of stacked objects classification.

**Table 1.** Accuracy performance in stacked objects classification.

| Method | Accuracy |
| :---: | :---: |
| PointConv | 87.1% |
| **SOCN** | **90.3**% |

### 5.2.2. Classification of Stacked Objects In Scene

In this part, the performance of our method is evaluated in a real scene. More than 10 objects were randomly stacked on the workbench. Considering the noise of external environment illumination and the change of object surface illumination caused by object stacking, we set the ambient light intensity to 100%, 50%, and 10%brightness to verify the robustness of our method. YOLO [33,34] is a state of the art method in object detection and it is widely employed in grasping systems [1,11–13]. We choose it along with PointConv as two main methods for the comparison against the proposed method in the grasping system. 1000 different placement scenarios of stacked objects are used to test SOCN under three different illumination conditions. Table 2 shows the classification results using different methods. It can be seen that at 100% brightness, the precision and recall rate of SOCN are slightly better than the YOLO method. By comparing the results between SOCN and PointConv, it can be seen that the performance of SOCN is better than PointConv in stacked object classification conditions due to the spatial scale and the self-attention mechanism. In addition, considering the illumination changes in stacked object recognition tasks, our method is more suitable than YOLO and PointConv. Figure 9 shows some cases of detection results using the proposed method, which accurately identifies cube objects marked with red boxes, pyramid objects marked with purple boxes, and cylinder objects marked with blue boxes under the 100%, 50%, and 10% brightness.

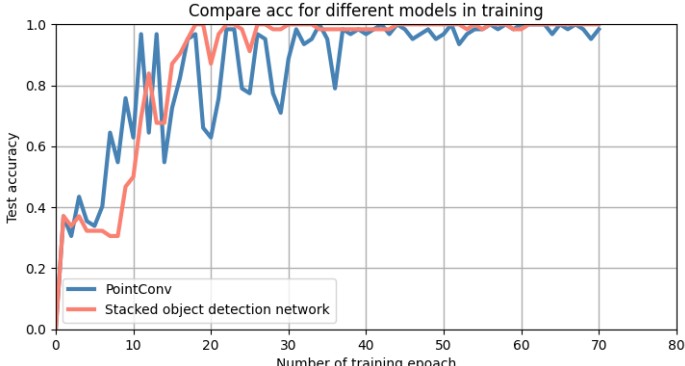

**Figure 8.** Training Performance. With the increasing number of epochs, the accuracy of SOCN becomes better and smoother as compared to PointConv.

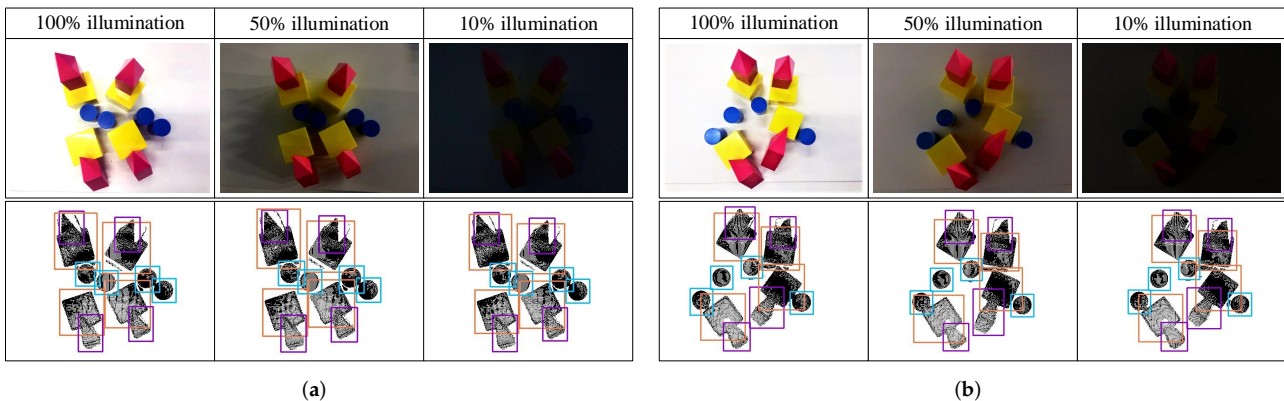

**Figure 9.** Stacked Object Classification using the proposed method. The red box represents the cube class, the purple box represents the pyramid class and the blue box represents the cylinder class. (**a**) Stacked object classification in the first scene; (**b**) Stacked object classification in the second scene.

**Table 2.** Performance in the stacked objects classification task.

| Method | 100% Illumination | | 50% Illumination | | 10% Illumination | |
|---|---|---|---|---|---|---|
| | Precision | Recall | Precision | Recall | Precision | Recall |
| **SODN** | **91.1**% | **87.6**% | **90.8**% | **87.9** | **90.6**% | **87.2**% |
| PoinConv | 84.6% | 85.2% | 83.3% | 85.7% | 83.8% | 84.4% |
| YOLO | 86.4% | 91.6% | – | – | – | – |

*5.3. Grasping Order Planning Results and Analysis*

The grasping order planning method is evaluated in multiple scenarios where objects are placed randomly and stacked on each other. By analyzing the process of the human body grasping objects, geometric relations and dependencies are the most important factor to avoid falling objects. In addition, people usually grasp attractive and easily broken objects first, and grab small objects before big ones. Based on this experience, the grasp coefficients are set as follows: The relation security scores are set as $RS_{descrete} = 3$, $RS_{top} = 2$, and $RS_{bottom} = 1$. There are three different categories of objects, and the class security scores of cube, cylinder and pyramid are set to 0.1, 0.2 and 0.3 respectively. Meanwhile, the size security scores are set as $SS_s = 0.03$, $SS_m = 0.02$, and $SS_l = 0.01$. The above security score parameters show that the geometric relations and dependencies are the most important factor in grasping tasks. Due to the class security scores are bigger than the size scores, the class factor is more important than the size factor in our experiment.

Figure 10 shows the experiment results in two grasping tasks. The classification result is displayed in the "Primitive" column, and the grasping order of our method planning is displayed in the "Order" column. Besides, the "Grasping" column shows the robot grasping processing. In both scenes, the robot first grasps the discrete object with the highest security score and then grabs the next object with the second-highest security score until all objects are grabbed. Besides, the grasping order of cube objects is behind cylindrical objects, which indicates that the planning of grasping order also considers the class and size of the objects.

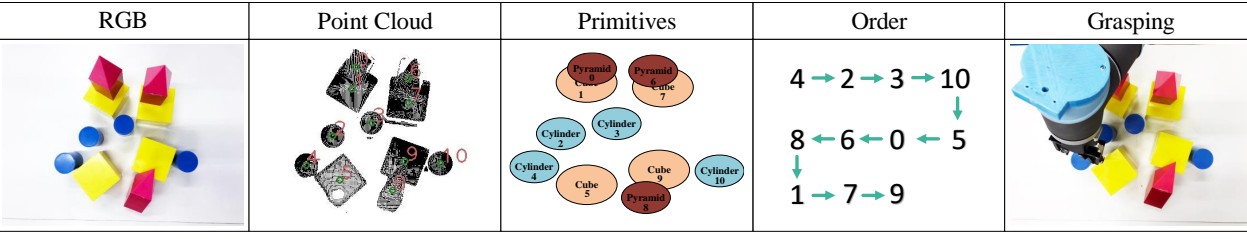

(**a**)

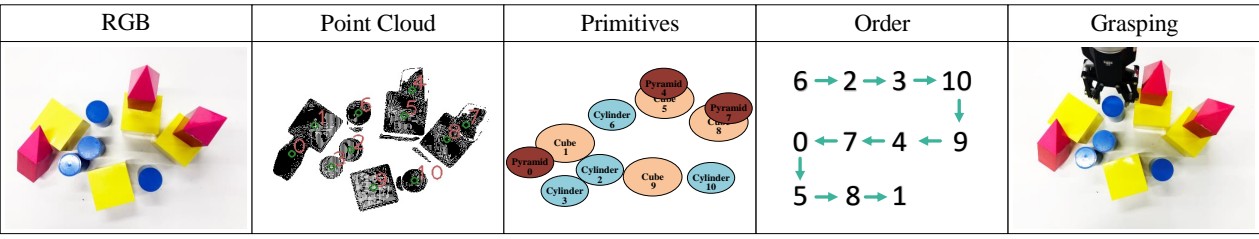

(**b**)

**Figure 10.** Grasping order planning test on stacked objects in two scenes. (**a**) The first scene; (**b**) The second scene.

## 6. Conclusions

This paper proposed a stacked object classification network and a grasping order planning method for robot grasping. SOCN can adjust the inverse density parameter by considering the spatial scale and has a self-attention mechanism to learn the important features of stacked objects. Because of these two properties, SOCN has high classification accuracy for stacked objects. On the other hand, the proposed grasping order planning method extracts the object relationships by using the designed relationship exploration network and calculates the security score to determine a safe grasping order. The experimental results validate the effectiveness of our proposed grasping method. In the future, we will optimize our grasping order planning method by considering more constraint conditions, like robot velocity constraint and trajectory constraint.

**Supplementary Materials:** There is the experiment video showing the performance of the proposed method. The video can be downloaded at https://www.mdpi.com/article/10.3390/electronics11050 706/s1.

**Author Contributions:** Conceptualization, W.L. and L.G.; Methodology, C.L. and D.J.; Software, D.J.; validation, C.L.; formal analysis, C.L. and D.J.; Writing—Original Draft Preparation, C.L. and D.J.; Writing—Review and Editing, W.L. and L.G.; Visualization, D.J.; Supervision, W.L. and L.G.; Project Administration, W.L. and L.G.; funding acquisition, W.L. and L.G. All authors have read and agreed to the published version of the manuscript.

**Funding:** This work was supported by the National Key R&D Program of China (No.2018YFB-1308400) and partially supported by the Portuguese Agency "Fundação para a Ciência e a Tecnologia" (FCT), in the framework of project UID/EEA/00066/2020.

**Acknowledgments:** The authors are thankful to the anonymous reviewers whose comments helped us to improve the paper.

**Conflicts of Interest:** The authors declare no conflict of interest.

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
