# Peer review of "Robot Grasping Based on Stacked Object Classification Network and Grasping Order Planning"

_electronics, doi:10.3390/electronics11050706_

Round 1

Reviewer 1 Report

This paper proposed the stacked object detection methods and grasping order planning based on the detection results. The stacked object detection method consisted of two parts which are the point convolution network to deal with point clouds and transformer to classify the object class. After the stacked object detection is done, the grasping order plan generated the secure grasping order for the robot arm. The proposed methods are interesting, but there are some points to be presented more clearly.

1) Authors said that they built their own dataset, but there are no details of dataset. They must provide the object classes, the number of data for each object class, sizes and heights of each object class, and so on. Also, they have to present how they build the dataset including more details. Perhaps, if authors open the dataset they built, then it will be great for other researchers.

2) Authors argued that the proposed SODN achieved good classification accuracy in Table 1 and Fig. 8. However, there is no information about the number of object classes, so it is hard to tell it is really good result or the dataset is too simple.

3) Authors compared the proposed method along with YOLO and PointConv for object detection. However, I don’t understand how the proposed SODN method can detect the object bounding box. Because, the proposed SODN only classifies the object class and there are no parts to predict object bounding box as shown in Fig. 3.

4) The proposed method name is stacked object detection network; however, the proposed network only classifies the object classes. Thus, it is not detection, it is stacked object classification network.

5) What is the meaning of lower parts of Fig. 9? Are they the detection results of the proposed method? Please add more explanations of figure.

6) Authors compared the proposed method along with YOLO and PointConv in different illuminations conditions. However, as I know, YOLO is the object detection algorithms based on the RGB colors and the proposed method is based on point clouds which are the depth information. The depth information is not influenced by illumination condition, on the other hands, the RGB information is really affected by illumination condition. Do authors think it is the fair comparison?

7) When I searched, there are several researches to deal with the stacked object detection; however, there are no surveys of these related research, so please add the related works for stacked object detection.

8) If it is possible, please provide the supplemental material for robot arm experiments such as experimental videos to show the proposed method performance more persuasively.

Author Response

We greatly appreciate the reviewers’ insightful and conscientious suggestions. We have revised our manuscript and done our utmost best to address all the comments and suggestions. All modifications are highlighted in yellow in the revised manuscript.
We also attach separated pdf file per received review with comments and clarifications.
Thank you.

Reviewer 2 Report

Besides, the results section can be improved. The comparison between methods using several light conditions (100%, 50%, 10%) is not a good test to show the accuracy of the methods in comparison to YOLO classifier. The depth sensor uses infrared light to estimate how far are the objects. The changes in the illumination in the scene do not introduce significant changes in the measured objects. This is the reason why the Precision and the Recall for the PoinConv and SODN do not have a significant change on those evaluation metrics. For me, it looks like YOLO is only taking the RGB image for the classification task. If not, please clarify this on this test. 

The attached PDF to this review only shows some of the typos in the document. Please change artefact to artifact in the entire document. 

Author Response

(The authors gave the same response as above.)

Reviewer 3 Report

Paper appears good and well built. The description of the used algorithm is clear as well as the experiments to validate it. I suggest to improve the references adding papers describing other grasping and classification techniques. See, for example, 

Riccardo Adamini, Nicholas Antonini, Alberto Borboni, Simone Medici,
Cristina Nuzzi, Roberto Pagani, Alberto Pezzaioli, Cesare Tonola,
User-friendly human-robot interaction based on voice commands and visual
systems, 24th International Conference on Mechatronics Technology (ICMT
2021), December 18th – 22nd, 2021 at Newcastle University in Singapore .

Author Response

(The authors gave the same response as above.)

Reviewer 4 Report

1:English writing and grammar need to be greatly improved. For example, there are a large number of first person subjects in the article.

2:The innovation points are not clear and need to be strengthened.

3: "a robot grasping methodology based on object detection and grasping order planning for stacked objects is studied." Reviewer  only saw the target identification and sequence, but did not see the so-called grasp.

4:There is no good comparative experiment

5:The author can win the opportunity of revision not because of your content, but because the reviewer thinks that the author is a "novice among scientific researchers", just want to give you a chance. Please cherish this opportunity of revision.

Author Response

(The authors gave the same response as above.)

Round 2

Reviewer 1 Report

Authors modified the manuscript based on the reviewer's comments well. For better presentation, this manuscript needs to be proofread by an expert. 

Reviewer 4 Report

The reviewer thinks it can be accepted and published.